# OCT Retinal and Choroidal Layer Instance Segmentation Using Mask R-CNN

**DOI:** 10.3390/s22052016

**Published:** 2022-03-04

**Authors:** Ignacio A. Viedma, David Alonso-Caneiro, Scott A. Read, Michael J. Collins

**Affiliations:** Contact Lens and Visual Optics Laboratory, School of Optometry and Vision Science, Queensland University of Technology (QUT), Kelvin Grove, QLD 4059, Australia; d.alonsocaneiro@qut.edu.au (D.A.-C.); sa.read@qut.edu.au (S.A.R.); m.collins@qut.edu.au (M.J.C.)

**Keywords:** deep learning, optical coherence tomography, region proposal, semantic segmentation

## Abstract

Optical coherence tomography (OCT) of the posterior segment of the eye provides high-resolution cross-sectional images that allow visualization of individual layers of the posterior eye tissue (the retina and choroid), facilitating the diagnosis and monitoring of ocular diseases and abnormalities. The manual analysis of retinal OCT images is a time-consuming task; therefore, the development of automatic image analysis methods is important for both research and clinical applications. In recent years, deep learning methods have emerged as an alternative method to perform this segmentation task. A large number of the proposed segmentation methods in the literature focus on the use of encoder–decoder architectures, such as U-Net, while other architectural modalities have not received as much attention. In this study, the application of an instance segmentation method based on region proposal architecture, called the Mask R-CNN, is explored in depth in the context of retinal OCT image segmentation. The importance of adequate hyper-parameter selection is examined, and the performance is compared with commonly used techniques. The Mask R-CNN provides a suitable method for the segmentation of OCT images with low segmentation boundary errors and high Dice coefficients, with segmentation performance comparable with the commonly used U-Net method. The Mask R-CNN has the advantage of a simpler extraction of the boundary positions, especially avoiding the need for a time-consuming graph search method to extract boundaries, which reduces the inference time by 2.5 times compared to U-Net, while segmenting seven retinal layers.

## 1. Introduction

Optical coherence tomography (OCT) imaging has become the standard clinical tool to image the posterior segment of the eye (i.e., the retina and choroid), since it provides fundamental information regarding the health of the eye [1]. The detailed high-resolution images allow clinicians and researchers to visualize the individual tissue layers of the posterior eye. These images are used to diagnose and monitor ocular diseases and abnormalities. Analysis of structural changes (thickness or area metrics) is commonly used as a surrogate of health status or disease progression [2]. In order to extract these metrics, tissue boundary positions need to be first segmented. Manual labelling of these boundaries requires experts to segment the areas of interest, which is a time consuming and subjective process, potentially prone to bias and errors [3,4,5].

Thus, the development of automatic methods of OCT image analysis (segmentation, classification) is fundamental to extract quantitative data from these medical images in a rapid and precise manner. Deep learning (DL), a sub-field of machine learning (ML), represents a new method to analyse these images with tools such as convolutional neural networks (CNN). Overall, CNN based methods have achieved remarkable performance in medical and natural image segmentation and are becoming the state of the art [6]. In retinal OCT image analysis, they have been used for a large range of applications such as full retinal segmentation [7], segmentation of retinal layers [8,9,10,11,12,13,14,15,16], intra-retinal fluid detection [17,18,19,20,21], and choroidal segmentation [22,23,24].

Depending on the type of network architecture, DL architectures for segmentation can be divided into different categories [25,26]. The most commonly used category in OCT retinal image segmentation is the encoder–decoder architecture, such as the well-known U-Net method [27], which has been extensively applied to OCT image analysis [4,15,19,20,21,28,29,30,31,32,33]. This architecture provides an output that represents a per-pixel probability map, which normally classifies a whole tissue region rather than a boundary of the tissue. Thus, from these output probability maps the boundaries of interest need to be extracted using an additional post-processing step such as graph search [10,34,35,36,37], or a second auxiliary neural network [14,21,30,38,39], which increases the processing time. Only a few of the proposed solutions provide a full end-to-end DL framework [15,40].

Interestingly, other newer DL architectures in the segmentation field have received significantly less attention for OCT image analysis. Among these DL architectures, the region proposal neural network architecture Mask R-CNN [41], has not been commonly applied to OCT image analysis. Of the region proposal architectures, Mask R-CNN differentiates from the others since it combines the object detection task with a second-stage of semantic segmentation in a process called instance segmentation [41]. This method addresses two of the main current limitations in the literature, (i) explores the application of region proposal method in OCT segmentation and (ii) provides an end-to-end solution to this application.

Figure 1a provides a graphical representation of the Mask R-CNN framework containing two main stages; the first stage being a Faster R-CNN architecture [42], which is graphically depicted in Figure 1b and can be further subdivided into three components: The backbone, the region proposal network and the object detection [41]. The first component (the backbone) consists of a CNN architecture used for image feature extraction and outputs the generated feature maps. These feature maps are then fed into the region proposal network (RPN), which is the second component and generates proposed bounding boxes (called anchors) for the object detection task, distributed over each feature map. These anchors are then classified by the RPN into two classes: Foreground class (positive anchors), which are anchors positioned in regions that represent features belonging to the objects to be detected, and background class (negative anchors) which are anchors positioned outside these objects. The positive anchors then go through a process called region of interest (ROI) alignment, centering them to the detected object and defining the ROIs to be used in the next module. The third and final component classifies the corresponding class inside each ROI for object detection. The second stage of the Mask R-CNN workflow consists of a fully convolutional mask prediction branch that performs the instance segmentation over each detected object in the image [41]. The advantage of instance segmentation is that it takes the information about the relative position of the retinal layers in OCT images into consideration before segmentation, allowing it to differentiate the instances of each class in the image and segment them individually.

This method has been used in different medical image segmentation applications including detecting and segmenting the optic nerve head structures in fundus images [43], detection and quantification of multiple retinal lesions in OCT images [44], and detection of retinal edema [45] in OCT images, but according to the studies found in the literature, its application to choroidal and retinal segmentation is still unexplored. The OCT retinal and choroidal layer segmentation field has been limited by a reliance on encoder–decoder DL architectures and a lack of end-to-end DL methods. This study aims to address these limitations by applying an alternative DL architecture (a region proposal method, that has attributes that may benefit more anatomically correct segmentation results) to OCT segmentation that is also capable of providing an end-to-end solution.

This study proposes the application and evaluation of a DL approach for the segmentation of OCT images of the posterior segment of the eye, incorporating a Mask R-CNN region proposal architecture, for posterior segment retinal and choroidal layers instance segmentation using OCT images from healthy eyes and comparing results with benchmark encoder–decoder methods. The contributions of this study are as follows:Evaluate the use of a region proposal architecture, based on the application of a Mask R-CNN method, as a new DL approach for retinal and choroidal layer instance segmentation using a large dataset of OCT images from healthy participants.The solution provides an end-to-end DL framework, which unlike most of the previously proposed methods eliminates the need for an additional post-processing steps to extract boundary positions. This reduces the inference time for retinal and choroidal layers boundary position segmentation.The proposed experiments cover some of the basic considerations on the use of the region proposal architecture for OCT image segmentation and explores some of the key network hyper-parameters to be considered to improve the performance when using Mask R-CNN for this approach.

This document is organized as follows: Section 2 presents more detail about the network and considered experiments. Section 3 presents the results and comparison with other methods together with a discussion of the main findings. Finally, Section 4 presents concluding remarks and suggests future work.

## 2. Materials and Methods

### 2.1. Data

A dataset consisting of spectral domain (SD) OCT images from healthy eyes was used for this study, with the data collection methods and population details described extensively elsewhere [46]. For completeness, only a brief summary is provided here. The data were captured using the Spectralis OCT instrument (Heidelberg Engineering, Heidelberg, Germany) from healthy participants (single instrument dataset). The original dataset contains images from over 100 children aged between 10 to 15 years (mean age, 13.1 ± 1.4 years) and each subject had data captured at four different visits, six months apart over an 18-month period. For each visit, two sets of six radial B-scans were captured on the same eye. To explore different image analysis tasks, two aspects of this dataset were used for the experiments, one included the retinal and intra-retinal layer segmentation, while the second included the global segmentation of the choroid and retinal layer structures. For the entire dataset, all boundaries were segmented automatically following a previously published work [47,48], after which an experienced trained observer reviewed and manually corrected any boundaries if needed. The observer’s manual segmentation has shown an excellent repeatability even for the complex task of choroidal segmentation, as reported in [46,49,50].

In the first analysis of the dataset, which was used to examine the intra-retinal layer segmentation, each image has seven retinal boundaries. The boundaries of interest can be visualized in Figure 2a. From the top of the image to the bottom, the layers include the internal limiting membrane (ILM), retinal nerve fibre layer (NFL), inner plexiform layer (IPL), outer plexiform layer (OPL), external limiting membrane (ELM), photoreceptor inner segment/outer segment junction layer (ISOS), and retinal pigment epithelium (RPE). For this experiment, the dataset was adapted using a total of 2135 OCT images obtained from 98 subjects belonging to visits 1 and 4 of the dataset. These images were split into three groups: The training set consisting of 1293 images belonging to 59 subjects; the validation set consisting of 429 images belonging to 20 subjects; and the evaluation set consisting of 413 images belonging to 19 subjects.

The second analysis of the dataset, which was used to explore the total retina and choroid segmentation task, contained the boundary information for three different layers. The boundaries can be visualized in Figure 2c. From the top of the image to the bottom, the layers include ILM, RPE boundary layers, as well as the choroid scleral interface (CSI). The full retina is encompassed between the ILM and RPE and the choroid lies between the RPE and CSI boundaries. Similar to the first dataset, the images were separated into three groups: The training set with 1311 images belonging to 60 subjects, a validation set with 431 images belonging to 21 subjects, and evaluation set with 406 images belonging to 18 subjects. All OCT images used in both datasets were taken from the first and last subject visits of the main dataset.

For both datasets, the data were separated to ensure different subjects were allocated in each set, thus avoiding mixing of information from the same subject into two different datasets. Thus, ensuring a fair assessment of the performance when evaluating the model, assessing only OCT images from new ‘unseen’ subjects. Before analysis, the dataset was adapted using the COCO dataset [51] format to use a structure that allows it to be easily processed by the Mask R-CNN algorithm. Figure 2 shows examples of the dataset and annotated masks, showing the defined masks used for the first (Figure 2b) and second (Figure 2d) analyses of the dataset.

### 2.2. Training

For the purpose of this experiment, the hyper-parameters were set following the default values defined in the Mask R-CNN study [41] with some modification. For training, a maximum of 1000 training epochs were defined with an automatic early stopping, which stopped the training when the model loss does not decrease after 75 consecutive epochs. Images were resized using a custom crop mode called “specific_crop”, which applies a 512×512 pixel size crop to the original OCT image. Given the dimensions of the OCT image (1534×512 pixels), five separate column wise fixed positions inside the image were defined (at pixel 0, 256, 512, 768, and 1022), the position expanded across the horizontal (transversal) dimension of the image to ensure the entire image is covered (fed to the network). The cropped sections have an overlap of 50% between adjacent sections. This option was preferred instead of using random coordinates, which may not ensure full image coverage. This configuration gives the square image dimensions needed to be processed by the model and ensures that the model will receive crops including all parts of the retina from the full OCT images and thus a more diverse training dataset.

The backbone CNN architecture used in this experiment corresponds to a ResNet50 with feature pyramid network (ResNet50-FPN) architecture proposed in [41]. This architecture combines the features extracted by the ResNet50 convolutional layers at different scales using the feature pyramid network method proposed in [52]. The total number of parameters for the Mask R-CNN model is equal to 44,698,312.

Unlike other datasets where this segmentation method can be applied, the OCT regions to be segmented do not have as much variation of size and ratio between the objects to be detected and to generate the bounding boxes. In other words, the anatomical dimensions of the retina provide a ‘natural’ constraint to the segmentation problem, thus modification of the size and ratio parameter was investigated to understand its impact on performance.

The RPN anchors were modified by only defining wide anchors that fit the specific retinal layer’s dimensions for the cropped OCT images. To understand the range of appropriate values, the ratios (width/height box dimension that fit the layer) of every retinal and choroid layer to be segmented in the cropped OCT images were analysed using the available data. Figure 3 shows the graphical analysis, where each histogram shows the frequency of the ratios per layer. The peak of each histogram is at a different location, ranging between 4 to 16. With 8 representing the most common ratio between the individual retinal layers. Due to the dimension reduction of the images after each convolutional block, four anchor ratios were defined for the first version of the dataset (16,32,64,128), to cover the feature maps obtained during the feature extraction process of the backbone. For the second version of the dataset, and after the same ratio analysis was performed, narrower ratios were used due to the sizes of the full retina and choroid tissues present in the OCT images (2,4,6,8,10). With the anchor ratios defined, five anchor scales were set (8,16,32,64,128) to fill the 512 pixels wide crops when using each ratio for the region proposals, these scales were kept the same for both datasets. The number of ROIs per image to be fed to the mask prediction branch was set to 200. For each training run of the model, data augmentation was used by applying rotation to each cropped image with a variation of −10/+10 degrees. All the segmentation methods were implemented in Python using Keras and Tensorflow (as the back end) modules, both of which are open-source software libraries for deep learning. The training process was performed on an Intel i7-9800X 3.80 GHz processor and Nvidia TITAN Xp GPU. Across the study, all the values represent the mean of 4 independent network runs, to ensure the proper assessment of the network repeatability and stability

### 2.3. Defined Tests

One of the appealing properties of DL models is their capability of being initialized with pre-trained parameters. The “pre-trained” model has been trained with different types of images from which they have learned features, which can benefit the new imaging analysis task. In general, the use of pre-trained models tend to provide an improvement in performance compared to those initialized with random weights. Additionally, the pre-trained model may require less training data to obtain good performance. Thus, the pre-trained model allows a reduction of computation time in the training of the model, in addition to providing a base on which it will begin its learning, improving its generalization ability [53].

To be able to understand how pre-training affects retinal layers segmentation performance, three types of simulations were conducted:Scratch, training a Mask R-CNN model from scratch without any previous learned feature, thus the network parameters are randomly initialized.COCO, initializing the Mask R-CNN model with the features learned using the COCO dataset [51].ImageNet, initializing the Mask R-CNN model with the features learned using the ImageNet dataset [54].

The use of the previously learned features in the initialization of the model (pre-trained model), also allows for different levels of fine-tuning. Thus, adjusting the parameters within feature extraction layers and adapting them to better solve the segmentation problem for the data set (OCT retinal images). Different convolutional blocks (i.e., level of the network) can be fine-tuned for different levels of abstraction, so another two simulations were defined when using the pre-trained models divided into four convolutional blocks. Three combinations were tested; training the features of all the convolutional blocks (called all), the last two convolutional blocks of the backbone (called 3+) and training the last convolutional block of the backbone (called 4+). After this test, the best performing model provides knowledge regarding which learned features are the best for initializing the model to learn the specific features for segmentation of retinal layers.

Finally, to put the findings of the Mask R-CNN into perspective, the performance was compared with results obtained using a state-of-the art U-Net approach described as “standard U-Net” in [23], which have been used previously for OCT retinal segmentation [7,15,36], which is a network that contains a total of 489,160 parameters. Additionally, a pretrained fully convolutional network (FCN) [55] and a DeeplabV3 [56] based on a ResNet50 architecture, which were adapted from [57], were used to compare results against a commonly used model with pretrained weights. The FCN model contains 25,557,032 parameters and DeeplabV3 39,635,014 parameters. The OCT images are not flattened or pre-processed prior to the segmentation tasks.

### 2.4. Performance Evaluation

To evaluate the method’s performance, two metrics were used, particularly the segmentation mask overlap, and boundary error metrics. The segmentation mask overlap metric, specifically the Dice overlap coefficient, has been used to report performance in different Mask R-CNN studies, while the mean (signed and absolute) boundary error, is a more clinically relevant metric to assess OCT segmentation performance, since the boundary error is closely related to thickness measurements which are commonly used for the quantification and clinical interpretation of OCT images. Similar to the training step, during the evaluation of the images, each image was cropped into five crops using the “specific_crop” method and each crop was segmented. After this, to have a fair assessment with other methods, which usually work on the entire OCT image, the five image crops with their corresponding segmentation information were fused to obtain the results for the whole OCT image. This is illustrated in Figure 4, where crops A, B, and C were used as main crops, and D and E were the auxiliary crops. These auxiliary crops are used to replace the border pixels of the main crops where the segmentation was not complete. This was done by cropping the necessary pixels from the auxiliary crops and replacing them in the main crops, rebuilding the OCT image with the information of the complete image segmentation results.

For the analysis of the results with the different methods, each segmentation output was post-processed to remove the border information of the full OCT image, specifically 100 pixels from the left side of the image were excluded due to the poor quality (low contrast tissue information) sometimes evident in this region. Similarly, on the right side of the image, 300 pixels were removed to mitigate the presence on the optic nerve head, which is an anatomical landmark that does not contain the boundaries of interest.

For the first performance analysis, the segmentation output of each network was analysed, by calculating the Dice coefficient corresponding to the overlap of each segmented retinal layer compared with its annotations. Then, for the boundary error analysis, the boundaries for the Mask R-CNN and FCN methods were directly obtained from the same segmentation outputs, by taking the upper and lower pixels belonging to each retinal layer, and internal layers with common boundaries, calculating the middle point between adjacent masks. For the U-Net and DeepLabV3 analysis, the boundaries were obtained using a graph-search method [23]. For these two methods (U-Net and DeepLabV3), direct boundary extraction yielded high boundary errors, thus the difference in techniques to post-process the boundaries. These data were used to calculate the mean signed and absolute boundary errors and the results between methods were compared.

## 3. Results

### 3.1. Retinal Layers Analysis

Initially, the performance comparison of different model initializations was carried out, as this provides the optimal model to be later used for comparison with the benchmark method. Table 1 shows the obtained results (Dice coefficient for each retinal layer) between the Mask R-CNN model trained from scratch, and the pre-trained models (COCO and ImageNet) used for initialization, and also the comparison of the different fine-tuned blocks.

A noticeable difference in performance for each initialization of the Mask R-CNN model can be observed. Using a model trained from scratch with randomly initialized parameters provides the worst segmentation performance when compared to using a model that has the same hyper-parameters, but pre-loaded weight values. This means that, using randomly initialized parameters may require a more detailed analysis of the hyper-parameters, a bigger volume of data and/or will need to be trained for longer to obtain comparable performance. However, the problem is solved when using the pre-trained models, which give better performance (i.e., higher Dice metrics for each individual retinal layer and the overall). When comparing between the different pre-trained models, the overall Dice values are close to each other, but the best performance is obtained when using the COCO dataset with an overall Dice coefficient of 94.13% across all considered layers. This indicates that the COCO dataset presents features that may be more suitable for the segmentation of retinal OCT images. Moreover, the best results belong to the tuning of the last two convolutional blocks of the backbone (3+), indicating that the low-level features (e.g., ridges, lines, spots) of the pretrained model contribute to improved performance and only the high-level features need to be updated.

After obtaining the best Mask R-CNN initialization for retinal layer segmentation, the effect of the anchor ratios on the network performance was investigated. The network’s default configuration for anchor ratios and sizes was compared to the defined configuration after the analysis performed on the size of the masks (Figure 3). Table 2 presents the results obtained from this analysis using Dice coefficients to compare performance when using the default (0.5, 1, 2) anchor ratios and anchor sizes (32, 64, 128, 256, 512) and the ratios and sizes proposed in Section 2.

When comparing the results, a substantial difference in segmentation performance can be observed, demonstrating the importance of performing a detailed analysis of the sizes and ratios of the objects to be detected before performing the segmentation. The overall performance for the network default ratios is 86.71%, versus the custom selected ratios with a 94.13%. The biggest difference corresponds to the ISOS-RPE layer with close to 40% difference, with a Dice coefficient of 54.83% for the default anchors, showing the importance of adequate selection of the hyper-parameters of the Mask R-CNN method. When performing a detailed analysis on the results, it was noticeable that low performance was associated with loss of retinal layers in the images that were not detected and thus not segmented.

Once the best initialization of the model, and highest performing anchor parameters were determined, an optimal Mask R-CNN was selected. The obtained results using the Mask R-CNN method shows good performance when segmenting each retinal layer, with an overall Dice coefficient of 94.13% and individual values ranging from 90.91% to 96.11%, demonstrating that the Mask R-CNN can be applied to the segmentation of retinal OCT images. To put these results into perspective, a U-Net, a FCN and a DeeplabV3 segmentation methods were used to perform the same segmentation task, using the same training and testing data. Table 3 shows the obtained results in terms of Dice coefficient for each retinal layer. This table compares the performance of the four methods for each retinal layer and the overall retina, before applying the post processing step for each method to extract the boundaries.

In the analysis performed when comparing the overall Mask R-CNN network with the reference models, results show only small differences. With the U-Net showing a slightly superior overall Dice coefficient (96.58%) compared to the Mask R-CNN (94.13%) and the DeeplabV3 showing a lower Dice coefficient (92.98%). Similar findings can be observed when analyzing each layer individually, with the U-Net showing the best performance in terms of Dice coefficient, except for the ILM-NFL which performs better using the Mask R-CNN method. The FCN and DeeplabV3 exhibited more variability across individual layers.

Figure 5 provides representative examples of the results for two images, all four methods show a very similar segmentation to the ground truth with the Mask R-CNN, FCN and DeeplabV3 demonstrating some pixelation of the mask around the centre of the OCT image, which corresponds to the foveal pit where the normal anatomical features of the retina mean that some of the retinal layers naturally disappear in this region.

A boundary error analysis was also performed. It is worth noting that boundary metrics, and the subsequently derived thickness metrics, are a more relevant clinical metric used to evaluate tissue health status. Thus, they may be considered potentially more relevant for assessing the performance of the methods. The corresponding retinal boundaries for the four methods were obtained from the segmented masks following the post-processing steps previously described (direct extraction for the Mask R-CNN, FCN and DeeplabV3, and graph-search for the U-Net). For boundary error analysis, results of mean signed and unsigned error were obtained. Table 4 shows the performance of all four methods focused on the segmentation of the boundaries.

Boundaries obtained from the Mask R-CNN segmented masks show an average of −0.087 pixels of mean signed error, and 0.784 pixels of mean absolute error, reinforcing the capability of this method to perform a retinal layer segmentation with a good level of performance. It is worth noting that the boundaries with Mask R-CNN, the FCN and DeeplabV3 were obtained without the use of any post-processing techniques, unlike the graph-search method used in the U-Net.

Boundary error results shows a close performance between methods. While assessing the average mean absolute error between these methods, the range of values between methods only varies 0.2 pixels, demonstrating the close overall agreement (similar segmentation performance).

A graphical comparison of the boundary, similar to the masks analysis, is shown in Figure 6. All four methods demonstrate good agreement with the ground truth, while delineating the retinal boundaries of interest. Regarding the inference time of each method when segmenting the retinal layers, Mask R-CNN method took an average of 3.25 s, the FCN method 2.87 s per segmented image, and the DeeplabV3 method 4.83 s per segmented image, with these three being end-to-end methods. While the U-Net together with graph-search post processing took in average 8.21 s per image. The time per image were obtained after averaging the time per image from every image analyzed across the four rounds.

### 3.2. Full Retina/Choroid Analysis

Similar to the previous section, the best initialization and hyper-parameters were set for training the Mask R-CNN method. In this, the second version of the dataset was used to analyse performance when segmenting the full retina and choroid regions of the OCT images. Table 5 compares the obtained Dice coefficient results for the segmentation of the full retina and choroid mask layers between Mask R-CNN, U-Net, FCN and DeeplabV3 methods.

When comparing the results obtained for the full retina and choroid regions, similar to the results obtained for the segmentation of the different retinal layers, the U-Net method presents a slightly superior performance in terms of Dice coefficient. However, the results show a close agreement. With respect to the U-Net, a difference of 2.29% and 1.28% in the retinal and choroidal regions was found compared to the Mask R-CNN, and less than 1% in both areas compared to the FCN and DeeplabV3 methods. When analysing the graphical results of each method (Figure 7), it is worth noting that the pixelation problem presented around the foveal region evident in the retinal layer analysis, is not affecting the results of the full retinal region segmentation.

The same post processing steps were performed for this segmentation approach to obtain the boundaries of interest: Extracting the ILM, RPE and CSI boundaries from the segmentation outputs of the four methods. Table 6 shows the boundary error analysis between the two methods for the segmentation of boundaries after the post-processing steps.

When analysing the boundary error results for the ILM, RPE and CSI layer boundaries, higher errors are found when using the Mask R-CNN method. From the graphical analysis shown in Figure 7 minimal difference is observed between methods, but it is possible to note “softer” curves when using the U-Net method for the full retina region segmentation.

For the segmentation of the full retina and choroid section, the Mask R-CNN method took an average of 1.91 s per segmented image, while the FCN took 1.83 s and the DeeplabV3 took 2.89 s. The U-Net with graph search post processing took an average 8.82 s per image.

As an addition to the Dice and boundary error, a number of segmentation metrics were calculated and analyzed to further compare the performance between methods. Table 7 shows the overall performance comparison obtained for the retinal layers and retina/choroid segmentation. From these results, the pixel accuracy, precision, recall, and specificity showed comparable values between methods, with only the recall showing a larger difference of 4.02% between Mask R-CNN and U-Net methods. It is worth noting that the DeeplabV3 provides the most consistent performance across both datasets while compared to the other methods, yet the boundary error metrics for the DeeplabV3 method do not show the same level of superior performance. The detailed performance obtained for these metrics are shown as Appendix Table A1, Table A2, Table A3, Table A4, Table A5, Table A6, Table A7 and Table A8.

### 3.3. Cross-Dataset Analysis

A final analysis was performed to assess the method’s performance on a different dataset, with a particular interest in assessing cross-dataset performance, thus providing a better understanding of the generalization of the model [58]. For this purpose, a publicly available age-related macular degeneration (AMD) dataset [59] was used.

The original AMD dataset contains images from 269 patients with AMD, and each patient had a set of volumetric scans composed of 100 OCT images (512 × 512 pixels). The scans were captured using the Bioptigen SD-OCT instrument, which is a different OCT instrument from the one used in the previous dataset (Spectralis SD-OCT). Each image in this AMD dataset corresponds to a single capture, without using any image averaging techniques, thus presenting more speckle noise than the original dataset (O.D.). In order to compare the performance with the O.D., two boundaries were used, corresponding to the ILM and RPE boundaries, which provides the full retinal tissue thickness as shown in Figure 2.

An initial test of the models was performed with a model trained on the O.D. (healthy dataset imaged with the Spectralis instrument) and testing on the AMD dataset (pathology dataset imaged with Bioptigen instrument). However, for all networks this method was not able to properly identify the classes of interest. Given the obvious differences between the dataset characteristics (presence of noise, image appearance and pathology) this is not an unexpected result.

To improve the segmentation performance, a small portion of images from the AMD dataset was added to the training of each model, adding a total of 756 images from 28 subjects (different from the images used for testing) to the original training O.D. dataset. Thus, providing the model an opportunity to learn from both datasets. Table 8 describes the Dice coefficient obtained when segmenting the retinal area, while Table 9 presents the boundary errors from the segmentation outputs.

While training with the combined dataset (O.D. + AMD), and testing on the O.D. only, most models present a comparable performance with minimal to no changes in the Dice or boundary error metric when compared to the original values (O.D./O.D.). Some marginal improvements can be observed in some of the metrics for some networks, this could be associated with the noise in the AMD dataset that could act as a regulation to the network training. While using the same training dataset (O.D. + AMD), but testing on the AMD exclusively, all models present a good performance with comparable metrics to the O.D. and show good Dice and boundary error metrics. However, the Mask R-CNN method seems to provide lower performance compared to the other models. This could be attributed to the difference in the dimension of the AMD dataset images (512 × 512 pixels) that does not allow the full segmentation processing to be performed as the O.D., which blends the segmentation results from different crops (as shown in Figure 4).

### 3.4. Discussion

When evaluating the use of the Mask R-CNN region proposal architecture, the pre-trained models improve the segmentation results by providing a basis on which the model to be trained using the most relevant features for OCT layer segmentation. Similar to the Mask R-CNN, the FCN and DeeplabV3 methods use pre-trained weights. However, in this study the proposed model provided lower performance than the other architectures, it is possible the network architecture with a large number of parameters or the initial object detection step included in the Mask R-CNN could underlie this difference in performance. In contrast, the U-Net model is not normally used with pre-trained weights. Most of the previous OCT segmentation studies have used a U-Net trained with randomly initialized parameters, which is the same approach used in our current work. The U-Net architecture was proposed with the objective of being able to perform well even when the model is randomly initialized and trained with low quantities of medical images [27]. Given the larger number of parameters of the Mask R-CNN network, the benefit seen from using a pre-trained model is not surprising.

One of the hyper-parameters found to be particularly important when using the Mask R-CNN method corresponds to the sizes and ratios of the anchors, with changes to these parameters resulting in a significant improvement in performance. The use of appropriate anchor sizes and ratios allows the trained model to detect objects that belong to those specific sizes and ratios in order to be later segmented.

When comparing with the U-Net, the segmentation results showed slightly better performance for the U-Net method with Dice coefficients of 2.45% and 1.80% in the overall metric for the retinal layer and choroidal datasets respectively. Interestingly while evaluating the boundary error, which is a more clinically relevant metric, the difference is minimal between methods, with a difference of mean absolute error of 0.12 and 0.15 pixels in the overall metric for the retinal layer and choroidal datasets respectively. It is worth noting the retinal layers are small in area, so changes of a few pixels may have a large impact on the Dice metric, yet these differences are less significant while assessing the boundaries. Similarly, when performing the analysis of the full retina versus the choroid region, higher Dice coefficients were obtained in the retina dataset for all methods. Similarly, the boundary error metrics showed little to no difference between methods for segmenting the retina and choroid.

While performing a graphical analysis of the segmentation results, it was also noted that the Mask R-CNN, FCN and DeeplabV3 methods can pixelate the outcome around the more curved zone of the foveal pit region. Outside the centre of the image, which has minimal curvature throughout the dataset, the method provided smoother segmentation results. Interestingly, this effect cannot be appreciated in the second dataset, so it is likely that the interaction between the retinal layers may play a role in this effect and a post-processing step may be needed to smooth this profile. When compared with U-Net results, it was possible to demonstrate one of the advantages of using the Mask R-CNN instance segmentation method. The U-Net is not capable of differentiating two different segmented portions as shown in Figure 5, where inside the ILM-NFL layer, there is a portion segmented as the ISOS-RPE class by the U-Net. This shows one of the advantages of the Mask R-CNN method, where the use of instance segmentation allows the method to differentiate both instances of the same segmented class (retinal layer) and to choose the one that corresponds to the correct position. This selection process may also be improved by applying anatomically correct segmentation, since the Mask R-CNN method has the ability to add the information on how classes (retinal layers) are connected to each other, and this positioning of the retina layers can be used in the training process using the “keypoints” feature [51].

The use of Mask R-CNN for retinal layer segmentation provides an end-to-end DL framework, which simplifies the post-processing required to extract the boundaries of interest. For example, the U-Net method used in this study for comparison requires a graph-search post processing step to obtain the boundaries. While the Mask R-CNN boundaries were obtained by taking the upper and lower pixel values of each segmented mask and calculating the middle point between those layers with common boundaries. This substantially simplifies the boundary extraction process and reduces the processing time.

## 4. Conclusions

While a large number of published methods have assessed OCT segmentation based on encoder–decoder architectures (U-Net and its variations), other DL architectures have received significantly less attention. In this paper, the Mask R-CNN method is applied to retinal and choroid layer segmentation. The proposed method provides good performance for segmentation of the retina (full and intra-layer) and choroid. This is particularly evident while assessing the boundary error metrics, which are the most clinically relevant performance indicator.

The promising results obtained from this study, demonstrate that the Mask R-CNN architecture can be used as an end-to-end method for OCT image segmentation. However, this experiment does not take full advantage of all the Mask R-CNN properties, so future work should consider hyper-parameters inside this method, anatomically correct segmentation using the instance segmentation and different losses. While the proposed Mask R-CNN model shows promising performance, the results, particularly those of the cross-dataset analysis, demonstrate that the performance may be linked to specific features of the OCT dataset. This cross-dataset analysis (different acquisition protocols, instruments and clinical features) and their effect on OCT image segmentation performance is an area that has received limited attention in OCT segmentation [34,60,61], and is thus worthy of further consideration.

## Figures and Tables

**Figure 1 sensors-22-02016-f001:**
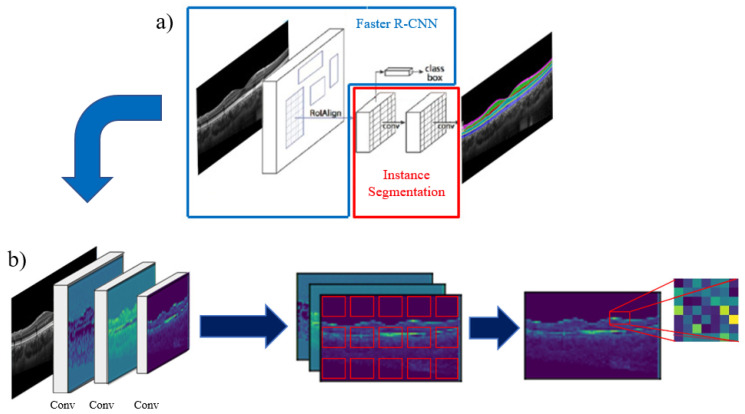
(**a**) Framework of the Mask R-CNN method used for retinal and choroid layer segmentation of OCT images. (**b**) Faster R-CNN workflow, receiving the image in the first stage to extract features at different scales, to then, generate anchors for object detection at each feature map, keeping the ones with more retinal layer information (non-background) and classifying the object inside the selected region of interest.

**Figure 2 sensors-22-02016-f002:**
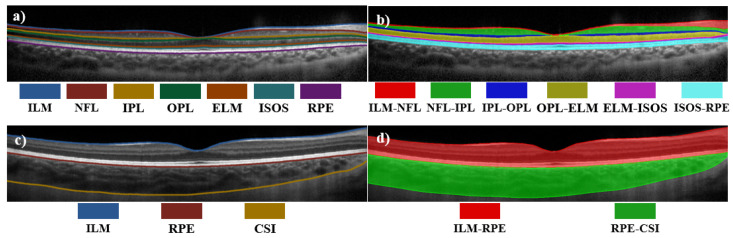
Example of an OCT image from the retinal dataset (**a**,**b**) and choroidal boundaries dataset (**c**,**d**). Left plots shows the boundaries of interest, while right plots show their corresponding class maps, which are extracted by the network.

**Figure 3 sensors-22-02016-f003:**
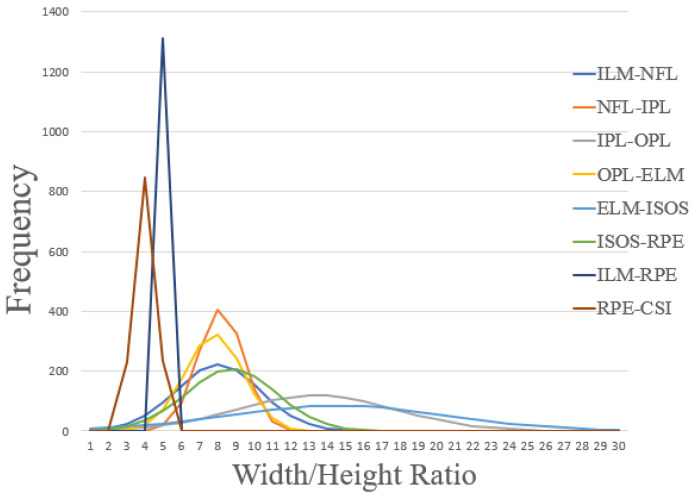
Retinal and choroid layer ratio (width/height) histograms for the dataset. Ratio is calculated from a frame (i.e., bounding box) which covers the maximum horizontal (width) and vertical (height) distances of each layer belonging to the raw OCT images without any pre-processing or flattening of the retinal area. The plots show the distribution of the ratios for each layer of the dataset and informed the selection of the anchor ratios hyper-parameter.

**Figure 4 sensors-22-02016-f004:**
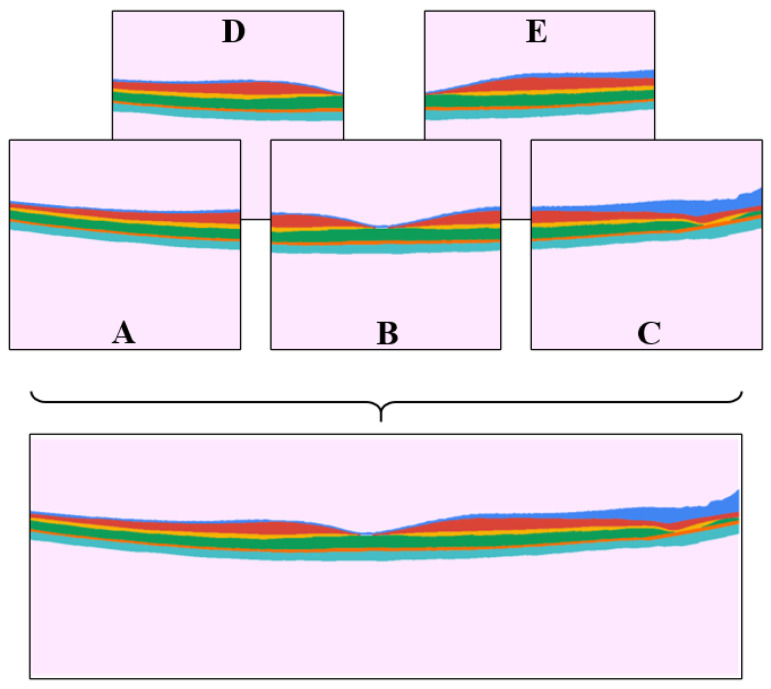
Example of the five crops obtained with “specific_crop” over an annotated OCT image used for training. The main crops (**A**–**C**) are joined with the support of the auxiliary crop (**D**,**E**) to produce a full image for evaluation performance.

**Figure 5 sensors-22-02016-f005:**
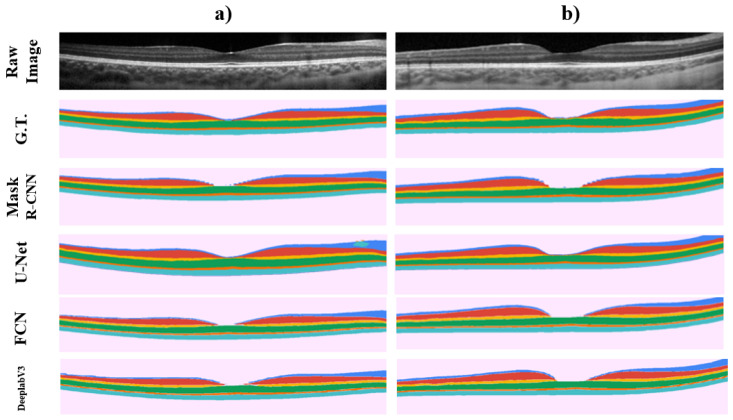
Graphical example of two different OCT images (Raw Image), showing their retinal layers ground truth (G.T.), and segmentation output probability maps when using the Mask R-CNN, U-Net, FCN and DeeplabV3 methods on each image respectively. Each row shows the proper segmentation map for the OCT image showed in columns (**a**,**b**).

**Figure 6 sensors-22-02016-f006:**
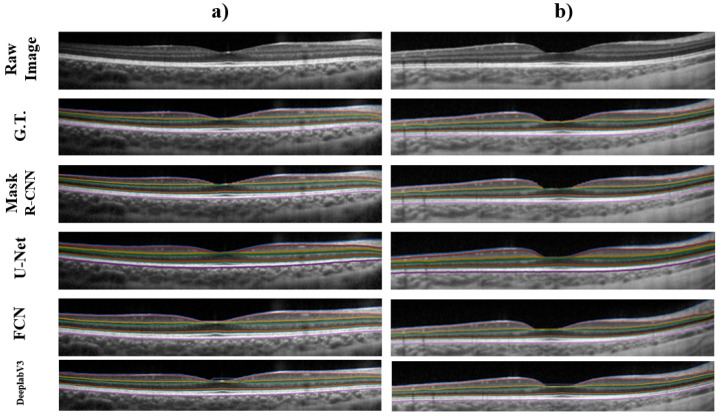
Graphical example of two different OCT images (Raw Image), showing their retinal layers ground truth boundary annotations (G.T.), and extracted boundaries from the obtained probability maps when using the Mask R-CNN, U-Net, FCN and DeeplabV3 methods on each image respectively. Each row shows the extracted boundaries for the OCT image showed in columns (**a**,**b**).

**Figure 7 sensors-22-02016-f007:**
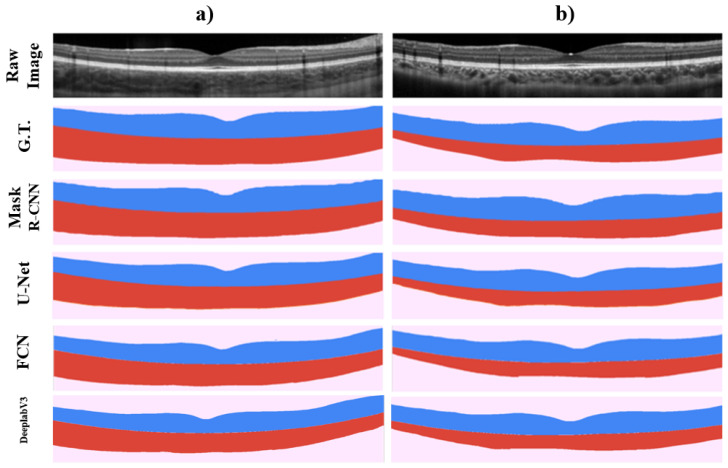
Graphical example of two different OCT images (Raw Image), showing their full retina and choroid tissue ground truth (G.T.), and segmentation output probability maps when using the Mask R-CNN, U-Net, FCN and DeeplabV3 methods on each image respectively. Each row shows the proper segmentation map for the OCT image showed in columns (**a**,**b**).

**Table 1 sensors-22-02016-t001:** Mean (standard deviation) Dice coefficient (%) for the individual retinal layer and overall performance for different model initializations. Each value represents the mean of four independent runs.

Init.	ILM-NFL	NFL-IPL	IPL-OPL	OPL-ELM	ELM-ISOS	ISOS-RPE	Overall
Scratch/All	94.85 (0.08)	91.04 (0.67)	87.78 (0.21)	88.27 (0.37)	87.50 (0.59)	88.90 (1.03)	92.03 (0.10)
COCO/All	95.82 (0.46)	92.09 (0.67)	90.56 (1.24)	90.61 (0.87)	90.37 (0.98)	91.89 (0.52)	93.70 (0.59)
COCO/3+	96.11 (0.04)	92.79 (0.34)	91.28 (0.24)	91.17 (0.41)	90.91 (0.26)	92.15 (0.20)	94.13 (0.09)
COCO/4+	95.46 (0.38)	91.93 (0.24)	89.72 (0.56)	90.17 (0.51)	89.21 (0.70)	90.16 (0.96)	93.14 (0.42)
ImageNet/All	95.73 (0.15)	92.02 (0.17)	89.36 (0.52)	89.58 (0.20)	89.96 (0.39)	91.43 (0.17)	93.32 (0.14)
ImageNet/3+	95.52 (0.28)	91.01 (1.27)	88.57 (1.70)	89.22 (0.81)	89.98 (0.74)	90.73 (1.39)	93.01 (0.68)
ImageNet/4+	95.39 (0.57)	91.01 (0.99)	88.58 (1.81)	88.57 (2.51)	88.89 (1.62)	90.11 (1.81)	92.68 (1.02)

**Table 2 sensors-22-02016-t002:** Mean (standard deviation) Dice coefficient (%) comparison between using the default and custom bounding boxes ratios and size for each individual and overall retinal layer. Each value represents the mean of four independent runs.

B.Box	ILM-NFL	NFL-IPL	IPL-OPL	OPL-ELM	ELM-ISOS	ISOS-RPE	Overall
Default	94.89 (0.22)	89.73 (0.63)	90.28 (0.54)	83.24 (2.51)	60.76 (8.28)	54.83 (7.36)	86.71 (1.55)
Custom	96.11 (0.04)	92.79 (0.34)	91.28 (0.24)	91.17 (0.41)	90.91 (0.26)	92.15 (0.20)	94.13 (0.09)

**Table 3 sensors-22-02016-t003:** Mean (standard deviation) Dice coefficient (%) of the Mask R-CNN (coco/3+ and custom ratios) compared with the benchmark U-Net method and a pretrained FCN and DeeplabV3 methods for each individual and overall retinal layer. Each value represents the mean of four independent runs.

Method	ILM-NFL	NFL-IPL	IPL-OPL	OPL-ELM	ELM-ISOS	ISOS-RPE	Overall
U-Net	94.32 (0.32)	96.74 (0.21)	93.44 (0.60)	97.84 (0.32)	94.78 (0.74)	98.06 (0.12)	96.58 (0.30)
Mask R-CNN	96.11 (0.04)	92.79 (0.34)	91.28 (0.24)	91.17 (0.41)	90.91 (0.26)	92.15 (0.20)	94.13 (0.09)
FCN	90.97 (0.12)	93.75 (0.11)	87.75 (0.04)	95.97 (0.02)	85.59 (0.06)	95.43 (0.01)	93.25 (0.04)
DeeplabV3	90.62 (0.14)	93.82 (0.09)	87.23 (0.19)	95.99 (0.05)	82.78 (0.18)	95.38 (0.05)	92.98 (0.01)

**Table 4 sensors-22-02016-t004:** Mean signed and absolute boundary errors (in pixels) from the retinal layer boundaries obtained after the corresponding post processing steps for Mask R-CNN, U-Net, FCN and DeeplabV3 methods. The average time of inference per image in seconds is presented. Each value represents the mean of four independent runs.

Method	ILM	NFL	IPL	OPL	ELM	ISOS	RPE	Average	Time/Img.
Mean signed error									
U-Net	−0.494 (0.067)	−0.408 (0.080)	−0.491 (0.050)	−0.623 (0.175)	−0.426 (0.165)	−0.500 (0.020)	−0.489 (0.117)	−0.490 (0.019)	8.21 (0.597)
Mask R-CNN	−0.136 (0.064)	−0.018 (0.127)	0.042 (0.043)	0.054 (0.086)	−0.031 (0.038)	0.042 (0.049)	−0.559 (0.154)	−0.087 (0.080)	3.25 (0.008)
FCN	0.492 (0.013)	0.368 (0.008)	0.500 (0.005)	−0.640 (0.044)	0.438 (0.019)	0.515 (0.025)	0.497 (0.016)	0.310 (0.017)	2.87 (0.048)
DeeplabV3	0.092 (0.022)	0.439 (0.020)	0.355 (0.040)	0.390 (0.035)	0.247 (0.025)	0.521 (0.027)	0.561 (0.020)	0.372 (0.025)	4.83 (0.353)
Mean absolute error									
U-Net	0.631 (0.040)	0.760 (0.006)	0.694 (0.032)	0.822 (0.127)	0.556 (0.059)	0.559 (0.015)	0.624 (0.084)	0.664 (0.029)	8.21 (0.597)
Mask R-CNN	0.969 (0.004)	0.962 (0.016)	0.817 (0.006)	0.814 (0.013)	0.548 (0.006)	0.513 (0.004)	0.862 (0.068)	0.784 (0.017)	3.25 (0.008)
FCN	0.839 (0.011)	0.860 (0.007)	0.904 (0.003)	0.971 (0.032)	0.616 (0.009)	0.719 (0.002)	0.744 (0.019)	0.808 (0.010)	2.87 (0.048)
DeeplabV3	1.082 (0.048)	1.060 (0.014)	0.873 (0.014)	0.797 (0.010)	0.634 (0.007)	0.680 (0.012)	0.732 (0.009)	0.837 (0.015)	4.83 (0.353)

**Table 5 sensors-22-02016-t005:** Mean (standard deviation) Dice coefficient (%) of the Mask -R-CNN (coco/3+ and custom ratios) compared with the benchmark U-Net method and a pretrained FCN and DeelpabV3 methods for the full retina (ILM-RPE) and choroid (RPE-CSI) regions segmentation. Each value represents the mean of four independent runs.

Method	ILM-RPE	RPE-CSI	Overall
U-Net	99.50 (0.01)	97.94 (0.07)	98.79 (0.03)
Mask R-CNN	97.21 (0.19)	96.66 (0.23)	96.99 (0.07)
FCN	98.86 (0.01)	97.72 (0.05)	98.35 (0.02)
DeeplabV3	98.87 (0.03)	97.79 (0.02)	98.46 (0.01)

**Table 6 sensors-22-02016-t006:** Mean signed and absolute boundary errors (in pixels) from the retinal and choroidal layer boundaries obtained after the corresponding post-processing steps for Mask R-CNN U-Net, FCN and DeeplabV3 methods. The average time of inference per image in seconds is presented. Each value represents the mean of four independent runs.

Method	ILM	RPE	CSI	Average	Time/Img.
Mean signed error					
U-Net	−0.456 (0.040)	−0.447 (0.073)	−0.861 (0.432)	−0.588 (0.163)	8.82 (0.622)
Mask R-CNN	−0.063 (0.133)	0.125 (0.119)	−0.219 (0.452)	−0.052 (0.183)	1.91 (0.010)
FCN	0.526 (0.036)	0.461 (0.276)	−0.381 (0.122)	0.202 (0.144)	1.83 (0.024)
DeeplabV3	0.527 (0.034)	0.376 (0.053)	0.362 (0.095)	0.422 (0.039)	2.89 (0.017)
Mean absolute error					
U-Net	0.601 (0.015)	0.596 (0.041)	2.539 (0.097)	1.245 (0.043)	8.82 (0.622)
Mask R-CNN	0.960 (0.017)	0.941 (0.017)	2.284 (0.081)	1.395 (0.037)	1.91 (0.010)
FCN	0.744 (0.015)	0.870 (0.050)	2.193 (0.069)	1.269 (0.044)	1.83 (0.024)
DeeplabV3	0.736 (0.016)	0.763 (0.020)	2.079 (0.023)	1.192 (0.016)	2.89 (0.017)

**Table 7 sensors-22-02016-t007:** Mean (standard deviation) pixel accuracy (Acc.), precision (Pr.), recall (Rc.), and specificity (Sp.) overall performance for the Mask R-CNN, U-Net, FCN and DeeplabV3 methods for the six retinal regions (ILM-RPE, NFL-IPL, IPL-OPL, OPL-ELM, ELM-ISOS, ISOS-RPE) and retina/choroid segmentation. Each value represents the mean of four independent runs.

Metric	Mask R-CNN	U-Net	FCN	DeeplabV3
Retinal regions				
Acc.	99.64 (0.01)	99.74 (0.01)	99.67 (0.01)	99.43 (0.01)
Pr.	94.22 (0.14)	95.60 (0.03)	93.14 (0.02)	98.23 (0.02)
Rc.	90.76 (0.13)	94.78 (0.05)	93.37 (0.04)	98.28 (0.01)
Sp.	99.87 (0.01)	99.88 (0.01)	99.83 (0.01)	99.66 (0.04)
Retina/Choroid				
Acc.	99.25 (0.04)	99.55 (0.01)	99.51 (0.01)	99.43 (0.01)
Pr.	99.08 (0.08)	98.51 (0.24)	98.27 (0.07)	99.34 (0.02)
Rc.	95.84 (0.29)	98.38 (0.20)	98.24 (0.03)	99.33 (0.01)
Sp.	99.85 (0.01)	99.75 (0.04)	99.70 (0.01)	99.51 (0.01)

**Table 8 sensors-22-02016-t008:** Cross-dataset analysis on the full retinal tissue segmentation results, presenting the mean (standard deviation) Dice coefficient (%) results. Three different training/testing combinations were included for each model, including: (i) The original using only the original dataset (O.D./O.D.), (ii) training adding the AMD dataset images (O.D. + AMD) and testing only on O.D. and (iii) training adding the AMD dataset images (O.D. + AMD) and testing only on AMD. O.D. denotes original healthy dataset imaged with the Spectralis device and the AMD dataset denotes the pathology dataset imaged with Bioptigen device.

Trained/Tested Data	Mask R-CNN	U-Net	FCN	DeeplabV3
O.D./O.D.	97.21 (0.19)	99.50 (0.01)	98.86 (0.01)	98.87 (0.03)
O.D. + AMD/O.D.	97.41 (0.26)	99.35 (0.01)	99.61 (0.01)	99.60 (0.01)
O.D. + AMD/AMD	96.40 (0.40)	99.55 (0.02)	99.83 (0.01)	99.83 (0.01)

**Table 9 sensors-22-02016-t009:** Cross-dataset analysis on the full retinal tissue segmentation results, presenting the mean (standard deviation) mean signed and absolute boundary errors (in pixels) results. Three different training/testing combinations were included for each model, including: (i) Using only the original dataset (O.D./O.D.), (ii) training adding the AMD dataset images (O.D. + AMD), and testing only on O.D. and (iii) training adding the AMD dataset images (O.D. + AMD), and testing only on AMD. The O.D. dataset denotes original healthy dataset imaged with the Spectralis device and the AMD dataset denotes the pathology dataset imaged with the Bioptigen device.

Data	Mask R-CNN		U-Net		FCN		DeeplabV3	
**Trained/ Tested**	**ILM**	**RPE**	**ILM**	**RPE**	**ILM**	**RPE**	**ILM**	**RPE**
Mean signed error								
O.D./O.D.	−0.063 (0.133)	0.125 (0.119)	−0.456 (0.040)	−0.447 (0.073)	0.526 (0.036)	0.461 (0.276)	0.527 (0.034)	0.376 (0.053)
O.D. + AMD/O.D.	0.451 (0.270)	−0.169 (0.010)	0.312 (0.007)	0.121 (0.102)	0.265 (0.148)	0.199 (0.103)	0.175 (0.084)	0.151 (0.074)
O.D. + AMD/AMD	1.299 (0.398)	−0.975 (0.810)	−0.034 (0.078)	0.030 (0.134)	−0.093 (0.032)	−0.140 (0.044)	0.035 (0.061)	−0.018 (0.140)
Mean absolute error								
O.D./O.D.	0.960 (0.017)	0.941 (0.017)	0.601 (0.015)	0.596 (0.041)	0.744 (0.015)	0.870 (0.050)	0.736 (0.016)	0.763 (0.020)
O.D. + AMD/O.D.	1.250 (0.316)	0.995 (0.187)	0.629 (0.002)	0.559 (0.052)	0.507 (0.060)	0.497 (0.037)	0.482 (0.020)	0.511 (0.026)
O.D. + AMD/AMD	1.887 (0.308)	2.434 (0.159)	0.343 (0.019)	0.408 (0.014)	0.387 (0.011)	0.523 (0.024)	0.421 (0.004)	0.557 (0.021)

## Data Availability

Not available.

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
