# Peer review of "OCT Retinal and Choroidal Layer Instance Segmentation Using Mask R-CNN"

_sensors, 2022, doi:10.3390/s22052016_

Round 1
Reviewer 1 Report
This study tested the feasibility of the Mask R-CNN architecture for OCT retinal layering, but I have the following suggestions about it.
1. The introduction to the development of machine learning in OCT is too lengthy and should be briefly summarised (lines 30 to 46)
2. The description of the CNN architecture is too much and could be reduced (lines 47-73)
3. The description of Mask R-CNN should cite the references (lines 75-101)
4. This study should additionally add a set of Mask R-CNN architectures that have adjusted hyperparameters and outperformed u-net in segmentation performance to increase the persuasiveness, which needs to be added.
Author Response
"Please see the attachment."

Reviewer 2 Report
The manuscript by Ignacio A. Viedma et al. demonstrated instance segmentation using Mask R-CNN for OCT retinal and choroidal images. The performance with quantitative metrics was also compared with the results obtained with other segmentation approaches, including standard U-Net, fully convolutional network, and DeeplabV3. Overall, the manuscript is well organized, and the results of this study should be of interest to the biomedical optics community. Specific comments are as follows:
- In lines 207 and 208, the authors mentioned that four anchor ratios were defined for the first version of the dataset (16, 32, 64, 128). How did the authors calculate these ratios according to the information from figure 3?
- Figure 3 shows retinal layer ratio (width/height) histograms for the datasets used in this study. How did the authors calculate the ratio for each layer? Before calculating the ratio, did the authors need another step to flatten each layer, which was also adopted in Ref. 51 proposed by Chiu et al.?
- In the paragraph of line 284, the authors mentioned: “boundaries for the Mask R-CNN method were directly obtained from the same segmentation outputs, by taking the upper and lower pixels belonging to each retinal layer, and for the layers with common boundaries, calculating the middle point between adjacent masks.” When segmenting OCT images using the fully convolutional network and DeeplabV3, did the authors use the same approach to find boundaries for each layer?
- From figure 1 of Ref. 23, the mask after segmentation using U-net could be obtained. Therefore, why cannot the authors use the aforementioned approach to find boundaries for each layer for U-Net analysis?
- Did the authors use the step about batch normalization for training the DL models, including Mask R-CNN, U-Net, fully convolutional network, and DeeplabV3?
- What kind of software did the authors use to train DL models and segment OCT data, for example, Python or Matlab?
Author Response
"Please see the attachment."

Reviewer 3 Report
The manuscript describes a Mask-RCNN approach for segmenting OCT retina and choroidal layers.
There are two main issues that make this manuscript not ready for publication:
- there are now a few open source datasets for OCT images, and the authors must include these datasets - I would suggest using the proprietary dataset and 1/2 open datasets for training and validation and then use a completely external dataset for testing as well
- the performance that the authors present with the Mask-RCNN method are lower than previous methods, albeit quicker. I don't think a few seconds quicker warrants lower segmentation performance.
Author Response
"Please see the attachment."

Round 2
Reviewer 1 Report
Modifications have been made by the authors as requested.
Author Response
Thank you
Reviewer 3 Report
I am not satisfied with the authors' response regarding the first point on open datasets. I believe that, at the very least, the authors should test their trained network on an open dataset and report the performances. This would help either justify their statement in the Conclusions or the statement that was added in this revised version could be slightly modified based on the reported performances.
Author Response
"Please see the attachment."
